∂ | **Open Peer Review** | Clinical Microbiology | Research Article

# Application of MALDI-TOF MS and machine learning for the detection of SARS-CoV-2 and non-SARS-CoV-2 respiratory infections

Sergey Yegorov,[1,2] Irina Kadyrova,[3] Ilya Korshukov,[3] Aidana Sultanbekova,[3] Yevgeniya Kolesnikova,[3] Valentina Barkhanskaya,[3] Tatiana Bashirova,[4] Yerzhan Zhunusov,[5] Yevgeniya Li,[5] Viktoriya Parakhina,[5,6] Svetlana Kolesnichenko,[3] Yeldar Baiken,[2,7,8] Bakhyt Matkarimov,[7] Dmitriy Vazenmiller,[3] Matthew S. Miller,[1] Gonzalo H. Hortelano,[2] Anar Turmukhambetova,[3] Antonella E. Chesca,[9] Dmitriy Babenko[3]

**ABSTRACT**  Matrix-assisted laser desorption/ionization time-of-flight mass spectrometry (MALDI-TOF MS) could aid the diagnosis of acute respiratory infections (ARIs) owing to its affordability and high-throughput capacity. MALDI-TOF MS has been proposed for use on commonly available respiratory samples, without specialized sample preparation, making this technology especially attractive for implementation in low-resource regions. Here, we assessed the utility of MALDI-TOF MS in differentiating severe acute respiratory syndrome coronavirus 2 (SARS-CoV-2) vs non-COVID acute respiratory infections (NCARIs) in a clinical lab setting in Kazakhstan. Nasopharyngeal swabs were collected from inpatients and outpatients with respiratory symptoms and from asymptomatic controls (ACs) in 2020–2022. PCR was used to differentiate SARS-CoV-2+ and NCARI cases. MALDI-TOF MS spectra were obtained for a total of 252 samples (115 SARS-CoV-2+, 98 NCARIs, and 39 ACs) without specialized sample preparation. In our first sub-analysis, we followed a published protocol for peak preprocessing and machine learning (ML), trained on publicly available spectra from South American SARS-CoV-2+ and NCARI samples. In our second sub-analysis, we trained ML models on a peak intensity matrix representative of both South American (SA) and Kazakhstan (Kaz) samples. Applying the established MALDI-TOF MS pipeline "as is" resulted in a high detection rate for SARS-CoV-2+ samples (91.0%), but low accuracy for NCARIs (48.0%) and ACs (67.0%) by the top-performing random forest model. After re-training of the ML algorithms on the SA-Kaz peak intensity matrix, the accuracy of detection by the top-performing support vector machine with radial basis function kernel model was at 88.0%, 95.0%, and 78% for the Kazakhstan SARS-CoV-2+, NCARI, and AC subjects, respectively, with a SARS-CoV-2 vs rest receiver operating characteristic area under the curve of 0.983 [0.958, 0.987]; a high differentiation accuracy was maintained for the South American SARS-CoV-2 and NCARIs. MALDI-TOF MS/ML is a feasible approach for the differentiation of ARI without specialized sample preparation. The implementation of MALDI-TOF MS/ML in a real clinical lab setting will necessitate continuous optimization to keep up with the rapidly evolving landscape of ARI.

**IMPORTANCE**  In this proof-of-concept study, the authors used matrix-assisted laser desorption/ionization time-of-flight mass spectrometry (MALDI-TOF MS) and machine learning (ML) to identify and distinguish acute respiratory infections (ARI) caused by SARS-CoV-2 versus other pathogens in low-resource clinical settings, without the need for specialized sample preparation. The ML models were trained on a varied collection of MALDI-TOF MS spectra from studies conducted in Kazakhstan and South America. Initially, the MALDI-TOF MS/ML pipeline, trained exclusively on South American

Address correspondence to Sergey Yegorov, yegorovs@mcmaster.ca, or Irina Kadyrova, ikadyrova@qmu.kz.

The authors declare no conflict of interest.

See the funding table on p. 10.

samples, exhibited diminished effectiveness in recognizing non-SARS-CoV-2 infections from Kazakhstan. Incorporation of spectral signatures from Kazakhstan substantially increased the accuracy of detection. These results underscore the potential of employing MALDI-TOF MS/ML in resource-constrained settings to augment current approaches for detecting and differentiating ARI.

**KEYWORDS** acute respiratory infection, COVID-19, SARS-CoV-2, MALDI-TOF MS, machine learning

The global response to the coronavirus disease 2019 (COVID-19) pandemic has underscored gaps existing in the laboratory-based diagnosis of acute respiratory infection (ARI) (1, 2). In the early stages of the pandemic, a shortage of rapid and inexpensive techniques amenable to modification to adapt to the newly characterized severe acute respiratory syndrome coronavirus 2 (SARS-CoV-2) motivated the search for alternative diagnostic tools. Matrix-assisted laser desorption/ionization time-of-flight mass spectrometry (MALDI-TOF MS), a technique traditionally employed in proteomics and metabolomics, has emerged as a promising alternative to molecular and immuno-chromatography-based assays to detect SARS-CoV-2 (3). Several different MALDI-TOF MS-based approaches involving varied degrees of sample preparation have been described (3).

Our clinical laboratory has been particularly interested in implementing a MALDI-TOF MS approach that could be used to assess ARI caused by both well-known and potentially novel pathogens. An approach that could address this is a MALDI-TOF MS pipeline that does not target specific structural components of pathogens but aims to characterize the host mucosal perturbations associated with respiratory infection. One such approach was developed in the early stages of the pandemic, applying machine learning (ML) algorithms to discern SARS-CoV-2-associated host mucosal changes using MALDI-TOF MS peak matrices acquired from nasopharyngeal swabs (NPS) without a specialized sample preparation (4–6). Therefore, in this study, we explored the feasibility and accuracy of this type of MALDI-TOF MS/ML in differentiating SARS-CoV-2 from non-COVID acute respiratory infections (NCARIs) in Kazakhstan.

## MATERIALS AND METHODS

### Study setting

We collected NPS from three participant subgroups: symptomatic SARS-CoV-2+, NCARIs, and asymptomatic controls (ACs). Two swabs were taken and then stored for subsequent SARS-CoV-2 PCR testing in DNA/RNA shield media (Zymo Research, Irvine, USA), and another in hosphate-buffered saline (PBS) (for MALDI-TOF MS analysis). Participants were recruited between 25 May 2020 and 20 December 2022. Written consent was obtained from all adult participants in the presence of a study coordinator; parental consent was obtained for participants under 18 years of age. The ARI diagnosis was made based on the presence of at least one of the following: fever, nasal congestion, cough, sore throat, and/or lymphadenopathy. SARS-CoV-2+ participants were recruited from patients of the Karaganda Regional Clinical Hospital, hospitalized with a PCR-confirmed SARS-CoV-2 infection. NCARI participants were recruited at the Karaganda Regional Clinical Hospital and the Karaganda City Centre for Primary Healthcare among patients admitted for moderate-severe ARI symptoms. Most (72.4%) NCARI participants were PCR-positive for common respiratory viruses (adenovirus, seasonal coronaviruses, bocavirus, parainfluenza viruses, respiratory syncytial virus, rhinovirus, influenza, or metapneumovirus) or bacteria (*Chlamydia pneumoniae* or *Mycoplasma pneumoniae*). Samples were collected around day 3 [median, interquartile range: IQR (2–4)] and day 5 [median, IQR: (3–7)] post-symptom onset for the SARS-CoV-2+ and NCARI participants, respectively. The AC sub-group was recruited from amidst the Karaganda University employees. The SARS-CoV-2 infection status was confirmed in the research lab for all samples using

SARS-CoV-2 PCR as described earlier (7–9). All samples were frozen and stored at −80°C until processing; samples underwent only one freeze/thaw cycle prior to MALDI-TOF MS.

In addition to the MALDI-TOF MS spectra obtained from clinical samples in Kazakhstan, we incorporated into our analysis the publicly available MALDI-TOF MS data from South America (4).

## MALDI-TOF MS analysis

Within feasible limits, we closely followed the published methodology for sample preparation, spectra acquisition, and preprocessing (4) with only minor modifications as specified. Spectral acquisition was performed on the MicroFlex LT v. 3.4 instrument (Bruker Daltonics, Bremen, Germany) equipped with a pulsed UV laser (N2 laser with 337 nm wavelength, 150 µJ pulse energy, 3 ns pulse width, and 20 Hz repetition rate). After thawing at room temperature, samples were spotted onto the steel target plate at 0.5 µL, covered with 0.5 µL of the alfa-Cyano-4-hydroxycinnamic acid (HCCA) matrix (a solution containing α-cyano-4-hydroxycinnamic acid diluted in acetonitrile, 2.5% trifluoroacetic acid, and nuclease-free water) and then air dried. The target plate was then loaded into the instrument. Spectra were generated by summing 500 single spectra (10 × 50 shots) in the range between 3 and 20 kDa, operating in positive-ion linear mode using an 18–20 kV acceleration voltage, by shooting the laser at random positions on the target spot.

## Spectral preprocessing

Raw MALDI-TOF MS files (Bruker) were uploaded and subsequently preprocessed in R (v. 4.3.0) using MALDIquantForeign and MALDIquant (10). To ensure consistency in peak processing with the original protocol, we used the R scripts generously shared by the authors (4). Briefly, the spectra were trimmed to a 3–15.5 kDa range, square-root transformed, and smoothened via the Savitzky–Golay method. Baseline correction was done using the TopHat algorithm, and intensity normalization was done via total ion current calibration as implemented in MALDIquant (10). Peak detection was performed using a signal-to-noise ratio of 2 and a half-window size of 10, and the peaks were binned with a tolerance of 0.003. Peak binning was performed in two stages to avoid any additional calibration differences. First, each group's spectra were binned separately, and peak filtration was performed, keeping only those peaks that were present in 80% of the spectra of each group. Subsequently, all peaks were binned together. The resulting peak intensity matrix was used for the downstream analyses.

In Analysis I, to assess the models trained on the South American samples from the source study (3), we made slight modifications to the sample preprocessing protocol as follows. To ensure that we are comparing the same 88 peaks, we employed the "reference" method for peak binning using the median values of the spectra and peaks obtained by Nachtigall et al. as a reference and eliminated the filtering procedure for each subgroup. In Analysis II, we constructed a *de novo* peak matrix representative of the combined South America and Kazakhstan data set using the script generously shared by Nachtigall et al.

## Principal component and hierarchical cluster analyses

Principal component analyses (PCAs) were performed using R FactoMineR (11) and factoextra (12). Dendrograms were generated using ggtree (13) and ggtreeExtra R (14). Hierarchical cluster analyses were done by calculating a distance matrix using the Euclidean method and clustering samples via the unweighted paired group with arithmetic mean method.

## Machine learning and statistical analysis

We implemented a total of seven ML algorithms, six of which were used in the earlier study (4) [DT (Decision Tree—Quinlan's C5.0 algorithm), KNN (k-Nearest Neighbors), NB

(Naive Bayes), RF (Random Forest), SVM-L (support vector machine with linear kernel), SVM-R (support vector machine with radial basis function kernel) plus XGBoost (eXtreme Gradient Boosting). Analysis I was executed by closely following the earlier protocol (4) with training performed on South American SARS-CoV-2+ and NCARI spectra. Since the training step of Analysis II incorporated three sub-groups (i.e., AC samples in addition to the SARS-CoV-2+ and NCARIs), the analysis pipeline was modified as outlined below to accommodate this change.

Initially, we split the entire sample into two distinct groups: the training data set, consisting of 80% of samples, and the test group, which accounted for the remaining 20%. In line with Nachtigall et al. (4), we conducted a training process using a fivefold (outer) nested repeated (five times) 10-fold (inner) cross-validation with a randomized stratified splitting approach. To optimize the performance of each algorithm, we tested 20 hyperparameters in the inner loop of the cross-validation approach, using a random search method. This process was repeated 20 times to ensure the robustness and reliability of the model. We selected the best models based on their area under the curve (AUC) score, which is a common metric for evaluating binary-classification model performance, using the Caret R package (15). In addition, model performance was assessed using several other classification metrics, including F-measure, recall, accuracy, specificity, sensitivity, and positive and negative predictive values in the yardstick R package; differences across the sub-groups were assessed using the Mann-Whitney *U* non-parametric test in R.

## RESULTS

The primary objective of the study was to assess the capacity of the MALDI-TOF MS approach to detect SARS-CoV-2 infection within a heterogeneous mix of SARS-CoV-2+, NCARI, and AC samples (Table 1).

Therefore, we performed two independent analyses (Fig. 1). In the first analysis, we assessed the performance of the Nachtigall et al. ML pipeline (4) on the combined pool of samples, both from the original study [data collected from three South American countries in 2020 (4)] and Kazakhstan (data collected in 2021 and 2022); the ML pipeline in this analysis was trained only on the original South American data sets. In the second analysis, we re-trained the ML algorithm, accounting for the spectra contributed by the samples from Kazakhstan and applied this re-trained ML algorithm to the combined pool of samples.

### Analysis I: applying the "as is" MALDI-TOF MS pipeline to differentiate ARI samples collected in Kazakhstan

To assess how well the original analysis pipeline (3) would differentiate SARS-CoV-2+ samples within the data set from Kazakhstan, we replicated the steps for (i) MALDI-TOF MS peak selection, (ii) ML training, and (iii) ML assessment. Specifically, we focused on the same MALDI-TOF MS peaks that Nachtigall et al. (4) used in their analyses (Table S1). These peaks were derived using a six-step spectra processing workflow including spectra transformation and smoothing, baseline removal, spectra calibration, peak detection, and peak processing.

We then constructed a peak intensity matrix on the 88 peaks, identical to that used by Nachtigall and colleagues (4), for the downstream analysis of a combined data set

**TABLE 1** Demographic characteristics of participants. AC: asymptomatic controls; NCARI: non-COVID acute respiratory infections.

| Characteristic | Overall, *N* = 252 | SARS-COV-2+ 2021, *N* = 108 | SARS-COV-2+ 2022, *N* = 7 | NCARI, *N* = 98 | AC, *N* = 39 | *P* value[a] |
|---|---|---|---|---|---|---|
| Age, years, median (IQR) | 38.0 (18.0, 60.0) | 61.0 (48.0, 69.0) | 3 (1.0, 37.0) | 8.0 (2.0, 35.0) | 34.0 (25.0, 47.0) | <0.001 |
| Male sex, *n* (%) | 114 (45.2%) | 49 (45.4%) | 6 (85.7%) | 38 (38.8%) | 21 (53.8%) | <0.001 |
| Kazakh ethnicity, *n* (%) | 116 (46%) | 26 (24.1%) | 5 (71.4%) | 61 (62.2%) | 24 (61.5%) | <0.001 |
| Any comorbidities | 104 (41.2%) | 72 (66.6%) | 1 (14.2%) | 15 (15.3%) | 14 (35.9%) | <0.001 |

[a]Differences across the groups were assessed using Kruskal-Wallis or Pearson's $\chi^2$ tests.

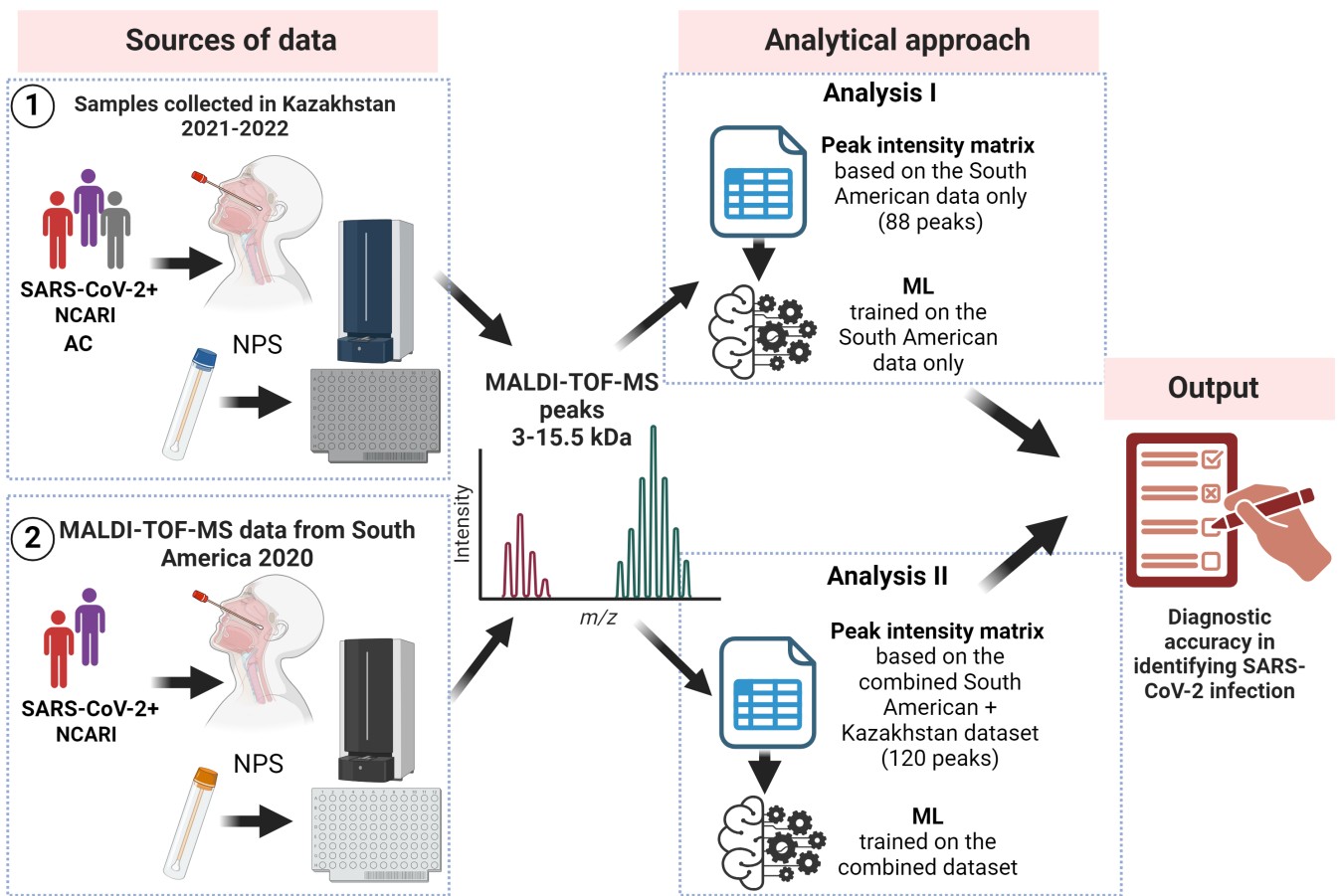

**FIG 1** Overall study workflow and description of the analyses. AC: asymptomatic controls; MALDI-TOF-MS: matrix-assisted laser desorption ionization mass spectrometry; ML: machine learning; NCARI: non-COVID acute respiratory infections; NPS: nasopharyngeal swab.

incorporating both the South American (Table S1) and Kazakhstan samples (Fig. 2; Table S2).

We next explored the selected peaks across the comparison groups by reducing the multidimensionality using principal component analyses and dendrograms. Like Nachtigall et al. (4), we did not detect any obvious clustering by sub-group, emphasizing the need for a more sensitive approach to discern subtle differences in the highly multidimensional MALDI-TOF MS peak data (Fig. 2D and E; Fig. S2 to S5). Hence, we then applied to our combined Kazakhstan-South America MALDI-TOF MS peak data set the original Nachtigall et al. ML algorithm trained on the original South American samples (4).

In keeping with earlier results (4), when tested the South American samples alone, SVM-R provided the highest receiver operating characteristic area under the curve (ROC AUC), although other models had similarly high-performance characteristics (Table S3; Fig. 3A and B) for classifying SARS-CoV-2 and non-SARS-CoV-2 cases.

Subsequently, we assessed the performance of the same ML algorithms on samples from Kazakhstan. Here, we observed a broad variation in the ability of the ML models to discern SARS-CoV-2+ samples. RF had the highest percentage of correctly identified 2020-SARS-CoV-2+ samples (91%) (Fig. 3A; Table S4 and Fig. S6). Notably, the accuracy for 2021 SARS-CoV-2 was <60% for all models, similar to the accuracy for identifying NCARI. RF discerned ACs with an accuracy of 68%, the highest of all models for this sub-group.

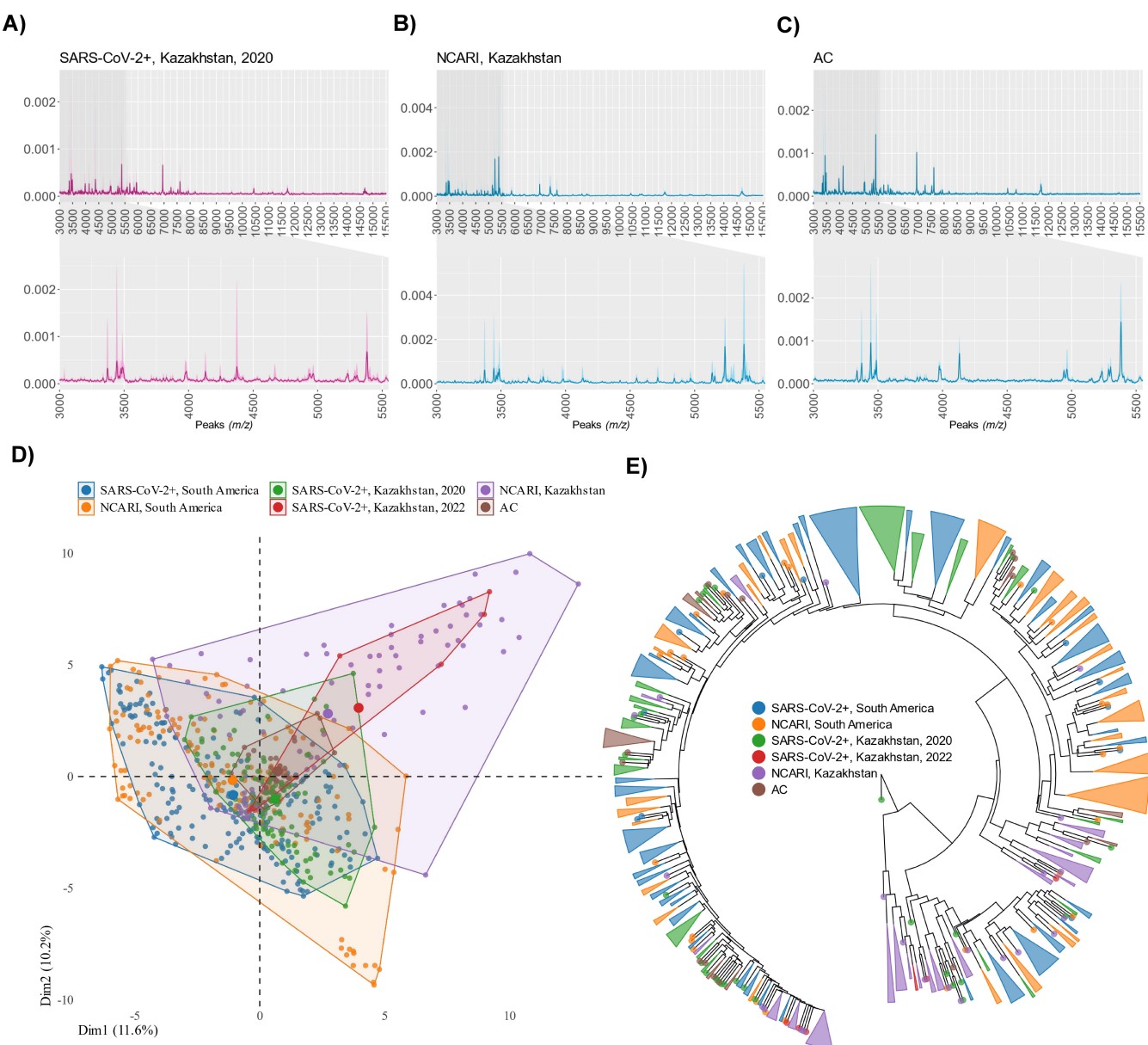

**FIG 2** MALDI-TOF MS peak data generated using nasopharyngeal swabs and processed following the MALDI-TOF MS/ML pipeline developed by Nachtigall and colleagues (4). (A–C) representative MALDI-TOF MS spectra from symptomatic SARS-CoV-2+ (A), symptomatic non-SARS-CoV-2 (B), and a healthy control sample from Kazakhstan (C). The central line indicates median value of the spectra, while the shaded region on either side represents the interquartile interval. Insets depict a range from 3,000 to 5,500 *m/z* encompassing 70% (62/88) of the identified peaks. (D) PCA of the combined data set incorporating MALDI-TOF MS data both from Kazakhstan and South America (2020 SARS-CoV+ and symptomatic SARS-CoV-2-negative) (3). (E) Dendrogram of the mass spectra stratified by sub-group from the combined dataset based on the peak intensity matrix for Analysis I.

## Analysis II: applying the re-trained MALDI/MS-ML to differentiate ARI

To ensure that we include all relevant MALDI-TOF MS signature peaks representative of all sub-groups, we performed peak selection on the entire pool of samples containing samples from both Kazakhstan and South America (*n* = 615). A total of 120 peaks were identified, and a peak intensity matrix was constructed (Table S5). As in Analysis I, PCAs and dendrograms did not show any visually apparent clustering of sub-groups (Fig. S7 to S10). We then proceeded to train ML models on the combined pool consisting of 120 peaks, of which 53 overlapped with the original 88 peaks.

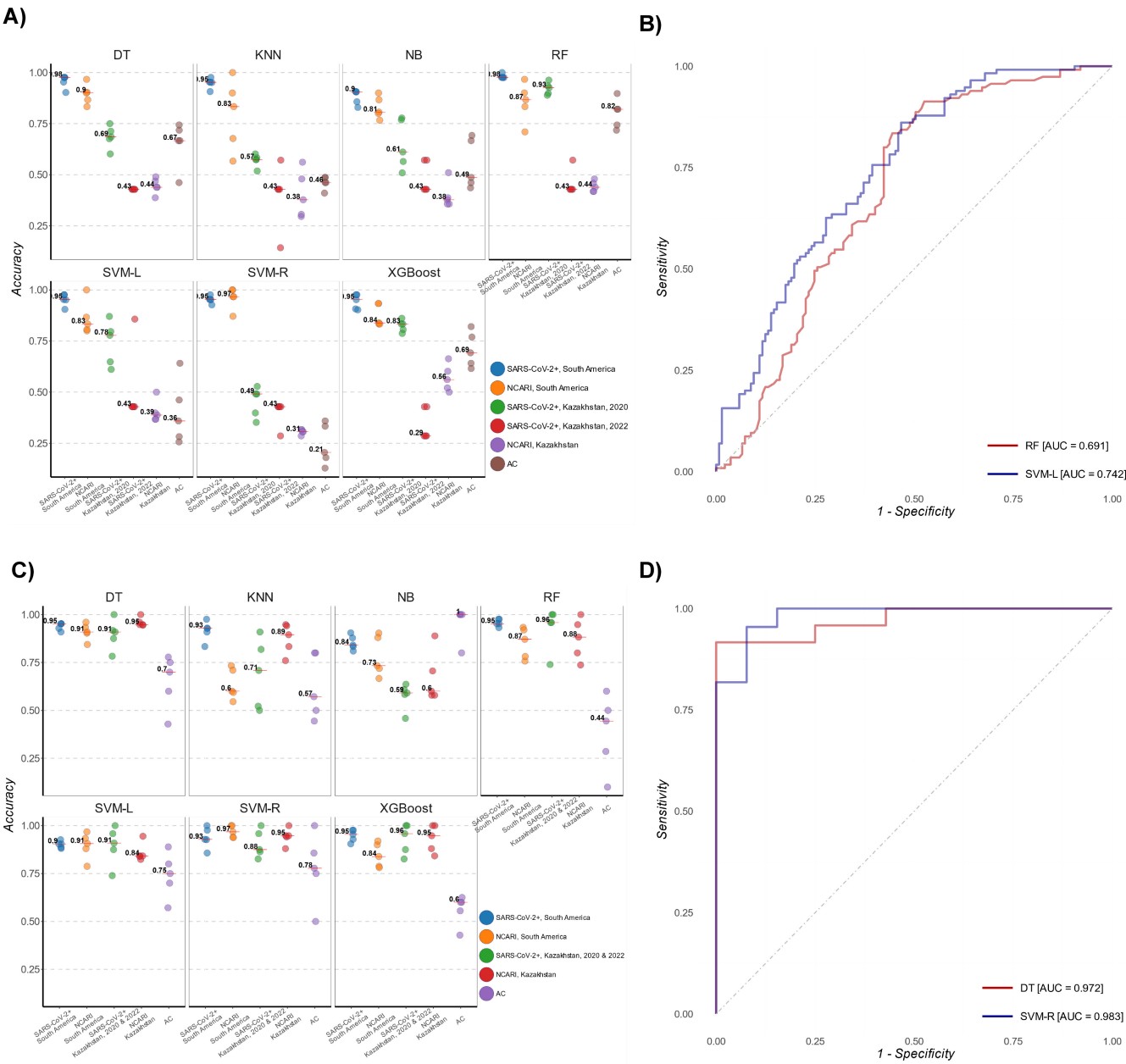

**FIG 3** Classification accuracy of the MALDI-ML algorithms assessed on the data from Kazakhstan and South America. (A) Accuracy metrics for each of the seven ML models trained on the South American MALDI-TOF MS data (Analysis I in the current study) for the differentiation of study sub-groups. (B) ROC curves of the top-performing RF and SVM-L algorithms (Analysis I). (C) Accuracy metrics for each of the seven ML models trained on the combined South America-Kazakhstan data set (Analysis II in the current study) for the differentiation of study sub-groups. (D) ROC curves for the top-performing SVM-R and DT algorithms (Analysis II).

Due to the small sample size of the 2022 subset, the 2021 and 2022 SARS-CoV-2+ subsets were combined prior to testing the model performance. We then assessed the performance of the trained ML algorithm on the South America-Kazakhstan data set. All models demonstrated similarly high-performance characteristics in differentiating SARS-CoV-2+ samples. SVM-R and DT slightly outperformed the other five models in discerning SARS-CoV-2 infection from both NCARIs and ACs with ROC AUC values of 0.983 [IQR: 0.958, 0.987] and 0.972 [IQR: 0.966, 0.979], respectively (Fig. 3C and D; Table S6). SVM-R, in particular, differentiated the Kazakhstan SARS-CoV-2+, NCARI, and AC subjects with an accuracy of 88.0%, 95.0%, and 78.0%, respectively (Fig. 3C). Both SVM-R

**TABLE 2** ROC AUC characteristics of RF and SVM-R models trained in Analysis I and Analysis II to differentiate SARS-CoV-2+ from both NCARI and AC samples within the Kazakhstan data set[a]

| AUC | Analysis I | | Analysis II | |
|---|---|---|---|---|
| | RF | SVM-R | RF | SVM-R |
| Median [IQR] | 0.67 [0.67; 0.68] | 0.60 [0.59; 0.63] | 0.93 [0.90; 0.96] | 0.98 [0.96; 0.99] |

[a]AUC are shown as medians and interquartile ranges.

and DT were also highly accurate at differentiating NCARI and AC sub-groups (Table S6). Table 2 summarizes the performance characteristics of RF and SVM-R for the differentiation of SARS-CoV-2+ samples in Analysis I vs Analysis II.

## DISCUSSION

Here, we aimed to assess the feasibility of deploying MALDI-TOF MS and ML in a clinical lab to differentiate SARS-CoV-2 from other ARIs, particularly in the context of minimal specialized sample preparation. Our initial application of the original MALDI-TOF MS/ML pipeline, trained on South American samples (4), demonstrated reduced efficiency in identifying samples from Kazakhstan. Re-training the ML models to incorporate MALDI peak information from a diverse pool of Kazakhstan samples, including SARS-CoV-2+, NCARI subjects, and asymptomatic controls, led to a substantial improvement in detection accuracy. Our results, viewed as proof-of-concept, support the feasibility of using MALDI-TOF MS combined with ML for identifying ARI in clinical laboratories with limited resources.

Our MALDI-TOF MS approach did not target specific viral structures but relied primarily on mass spectrometric signatures associated with ARI-induced perturbations in the nasal mucosa (3). Advantageously, this would potentially allow for the detection of a broad range of ARI without prior knowledge of specific viral markers or the need for specialized reagents or probes (which are often scarce in the early stages of epi/pandemics). On the other hand, the specificity of our approach would be affected by the overlap in mucosal immune signatures of different ARI pathogens and their sub-strains (16). Thus, although in theory, the current MALDI-TOF MS approach could more robustly accommodate emerging SARS-CoV-2 and other ARI variants compared to approaches that target specific viral proteins (17, 18), substantial changes in the host response to ARI could still impact the algorithm's accuracy.

Uniquely, in our analysis, we used publicly available MALDI-TOF MS data to expand and diversify our ML training, thus including MALDI-TOF MS data from South America and Kazakhstan. Unfortunately, our attempts were unsuccessful in obtaining MALDI-TOF MS data from researchers who used similar MALDI-TOF MS approaches but in Europe and North America (5, 6). Establishment of an open-access MALDI-TOF MS database could mitigate this barrier in the future and enable more rapid deployment and adaptation of the technology across different geographic areas and stages of epi/pandemics. This would allow the development of preliminary models that could be refined and improved as local data become available.

Our replication studies underscore the importance of considering geographical and population-specific variations in the application of MALDI-TOF MS/ML. The differences seen in Analysis I vs Analysis II in the performance of the original pipeline (trained solely on South American samples) may be attributed to the inherent complexity of NPS, which contains a mixture of host proteins and diverse microbial species and to the variability in immune response across different viral loads, SARS-CoV-2 variants, and co-infections (16, 19, 20). Thus, the improved performance of the algorithm re-trained in Analysis II could be due to both the expanded size and geographic diversity of the combined Kazakhstan-South America dataset. For example, a larger data set in Analysis II incorporating three participant groups (SARS-CoV-2, NCARIs, and ACs) supplied a wider spectrum of ARI-associated molecular signatures. At the same time, inclusion of MALDI-TOF MS data from such geographically distant regions contributed additional molecular signatures

underlined by the genetic, environmental, and behavioral differences between the cohorts in South America and Kazakhstan.

Our analysis has several limitations. The lack of specialized sample preparation, although advantageous for low-resource settings, may introduce variability and noise into the data, a concern raised by other authors (17, 18). Due to a relatively small sample size of the NCARI group, we did not further pursue stratification of this group by the causative agents identified via multiplex PCR. The utility of MALDI-TOF MS/ML in differentiating various NCARIs and unrecognized SARS-CoV-2 infections would be important to examine in the context of the changing post-pandemic ARI landscape (21–23). Temporal variation, spanning samples collected over 2 years (2020–2022), and demographic differences across groups might have contributed to a high heterogeneity of our results. To address these constraints within a clinical laboratory setting, it would be necessary to integrate the MALDI-TOF MS/ML pipeline into a comprehensive ARI testing strategy and train the pipeline on a sufficiently large pool of samples (2). The rate of pathogen emergence and geographical diversity will also dictate the need for algorithm recalibration, requiring continuous monitoring and integration of global ARI data. Further research is needed to explore the specific components of MALDI-TOF MS spectra that are most informative for differentiating various ARI.

In conclusion, our study provides insights into the potential of MALDI-TOF MS as an accessible laboratory-based diagnostic tool for ARI. While promising, the implementation of MALDI-TOF MS/ML in real clinical lab settings will require further optimization, validation, and continuous adaptation to the evolving epidemiological landscapes. We are hopeful that our study will encourage more transparency and data sharing in the field of clinical MALDI-TOF MS research.

## ACKNOWLEDGMENTS

We thank the study participants and clinic staff. We are grateful to Professor Leonardo Santos for sharing the R scripts and associated data from their original study (4). Figure 1 was created with BioRender.com.

Authors provide consent for the publication of the manuscript detailed above, including any accompanying images or data contained within the manuscript.

The funders had no role in study design, data collection and analysis, decision to publish, or preparation of the manuscript.

## AUTHOR AFFILIATIONS

[1]Department of Biochemistry and Biomedical Sciences, Michael G. DeGroote Institute for Infectious Disease Research, McMaster Immunology Research Centre, McMaster University, Hamilton, Ontario, Canada

[2]School of Sciences and Humanities, Nazarbayev University, Astana, Kazakhstan

[3]Research Centre, Karaganda Medical University, Karaganda, Kazakhstan

[4]City Centre for Primary Medical and Sanitary Care, Karaganda, Kazakhstan

[5]Infectious Disease Centre of the Karaganda Regional Clinical Hospital, Karaganda, Kazakhstan

[6]Department of Internal Diseases, Karaganda Medical University, Karaganda, Kazakhstan

[7]National Laboratory Astana, Centre for Life Sciences, Nazarbayev University, Astana, Kazakhstan

[8]School of Engineering and Digital Sciences, Nazarbayev University, Astana, Kazakhstan

[9]Faculty of Medicine, Transilvania University, Braşov, Romania

## AUTHOR ORCIDs

Sergey Yegorov  http://orcid.org/0000-0002-7136-7921
Irina Kadyrova  http://orcid.org/0000-0001-7173-3138
Aidana Sultanbekova  http://orcid.org/0000-0002-7136-7921

## FUNDING

| Funder | Grant(s) | Author(s) |
|---|---|---|
| Science Committee of the Ministry of Education and Science of the Republic of Kazakhstan | No.AP09259123 | Irina Kadyrova |
| Science Committee of the Ministry of Education and Science of the Republic of Kazakhstan | No. AP19679717 | Yeldar Baiken |
| Canadian Institutes of Health Research | Postdoctoral fellowship (Competition #202210MFE) | Sergey Yegorov |
| Faculty Development Competitive Research Grant (COVID) from Nazarbayev University | 280720FD1902 | Gonzalo H. Hortelano |

## AUTHOR CONTRIBUTIONS

Sergey Yegorov, Conceptualization, Data curation, Formal analysis, Methodology, Supervision, Validation, Visualization, Writing – original draft, Writing – review and editing, Investigation, Project administration | Irina Kadyrova, Conceptualization, Data curation, Formal analysis, Funding acquisition, Investigation, Project administration, Resources, Supervision, Writing – original draft, Writing – review and editing | Ilya Korshukov, Data curation, Formal analysis, Investigation, Writing – original draft, Writing – review and editing | Aidana Sultanbekova, Data curation, Investigation, Methodology, Validation, Writing – original draft, Writing – review and editing | Yevgeniya Kolesnikova, Data curation, Investigation, Methodology, Writing – review and editing | Valentina Barkhanskaya, Data curation, Investigation, Methodology, Writing – review and editing | Tatiana Bashirova, Investigation, Methodology, Resources, Writing – review and editing | Yerzhan Zhunusov, Data curation, Investigation, Methodology, Resources, Writing – review and editing | Yevgeniya Li, Data curation, Resources, Writing – review and editing | Viktoriya Parakhina, Investigation, Resources, Writing – review and editing | Svetlana Kolesnichenko, Conceptualization, Data curation, Investigation, Methodology, Writing – original draft, Writing – review and editing | Yeldar Baiken, Funding acquisition, Methodology, Writing – review and editing, Resources | Bakhyt Matkarimov, Methodology, Resources, Writing – review and editing, Supervision | Dmitriy Vazenmiller, Investigation, Methodology, Writing – review and editing | Matthew S. Miller, Supervision, Writing – review and editing, Conceptualization, Investigation | Gonzalo H. Hortelano, Funding acquisition, Investigation, Writing – review and editing, Conceptualization, Project administration, Resources, Supervision | Anar Turmukhambetova, Funding acquisition, Resources, Writing – review and editing | Antonella E. Chesca, Conceptualization, Data curation, Writing – review and editing | Dmitriy Babenko, Conceptualization, Data curation, Formal analysis, Investigation, Methodology, Software, Validation, Visualization, Writing – original draft, Writing – review and editing

## DATA AVAILABILITY

All raw data and R code are available through Github (https://github.com/dimbage/ML_MALDI-TOF_SARS-CoV-2).

## ETHICS APPROVAL

All study procedures were approved by the Research Ethics Board of Karaganda Medical University under Protocol #12 (approval #45) from 06.04.2020. Written informed consent was obtained from all participants.

## ADDITIONAL FILES

The following material is available online.

## Supplemental Material

**Supplemental material (Spectrum04068-23-s0001.pdf).** Figures S1 to S10; Tables S1 to S6.

## Open Peer Review

**PEER REVIEW HISTORY (review-history.pdf).** An accounting of the reviewer comments and feedback.

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
