## [Reviewer comments · Microbiology Spectrum]

Microbiology Spectrum

Application of MALDI-TOF MS and Machine Learning to Detection of SARS-CoV-2 and non-SARS-CoV-2 Respiratory Infections.

Sergey Yegorov, Irina Kadyrova, Ilya Korshukov, Aidana Sultanbekova, Yevgeniya Kolesnikova, Valentina Barkhanskaya, Tatiana Bashirova, Yerzhan Zhunusov, Yevgeniya Li, Viktoriya Parakhina, Svetlana Kolesnichenko, Yeldar Baiken, Bakhyt Matkarimov, Dmitriy Vazenmiller, Matthew Miller, Gonzalo Hortelano, Anar Turmuhambetova, Antonella Chesca, and Dmitriy Babenko

Corresponding Author(s): Irina Kadyrova, Karaganda Medical University

Review Timeline:

Submission Date:	December 1, 2023
Editorial Decision:	January 7, 2024
Revision Received:	February 22, 2024
Accepted:	February 28, 2024

Editor: Heba Mostafa

Reviewer(s): The reviewers have opted to remain anonymous.

Transaction Report:

DOI: <https://doi.org/10.1128/spectrum.04068-23>

Re: Spectrum04068-23 (Application of MALDI-MS and Machine Learning to Detection of SARS-CoV-2 and non-SARS-CoV-2 Respiratory Infections.)

Dear Dr. Irina Kadyrova:

Thank you for the privilege of reviewing your work. Below you will find my comments, instructions from the Spectrum editorial office, and the reviewer comments.

Revision Guidelines

Sincerely,
Heba Mostafa
Editor
Microbiology Spectrum

Reviewer #1 (Comments for the Author):

'Application of MALDI-MS and Machine Learning to Detection of SARS-CoV-2 and non-SARS-CoV-2 Respiratory Infections' by Yegorov, et al. is a study about the practical use of MALDI and machine learning to differentiate between respiratory infections caused by SARS-CoV-2 versus those that are not, with the potential to use this technology in a clinical setting.

Discussion points

- You point out that an advantage of this technique could be for limited resource labs in the early stages of a pandemic (Line 233), but how would that be possible without the large pool of samples to train the ML models on, and also with a standard of care test to show true positives?
- Line 236, 254 - you acknowledge the importance of taking geographical location of the sample into consideration, so how would this technology work in the clinical lab? Would there be other options for testing if someone was from or had traveled out of the region when infected? How often would the ML model need to be recalibrated to ensure it wouldn't miss mutations/new strains? Would it be part of a testing algorithm?

Line 73. Remove 'among'

Line 83. How long were samples frozen for before testing? Was there only one freeze/thaw?

Table 1. Discrepancies in how percentages are written - some with a '.' and some with a ',' - I would change the three percentages that have a ',' (SARS-CoV-2+ column) to having a '.' for consistency.

Reviewer #2 (Comments for the Author):

The authors present an interesting extension of prior MALDI-TOF-MS and Machine Learning (ML) methods for the identification of COVID-19 and other non-COVID respiratory viruses in Kazakhstan. This work specifically highlights the importance of training ML algorithms on geographically diverse datasets. Overall, the findings of the study are well supported. In the attached review are a few recommendations for expanded discussion, updates to figures/tables to improve clarity, and other minor changes.

The authors present an interesting extension of prior MALDI-TOF-MS and Machine Learning (ML) methods for the identification of COVID-19 and other non-COVID respiratory viruses in Kazakhstan. This work specifically highlights the importance of training ML algorithms on geographically diverse datasets. Overall, the findings of the study are well supported. Below are a few recommendations for expanded discussion, updates to figures/tables to improve clarity, and other minor changes.

Main points

- In the discussion, the authors may consider commenting on the potential impact of COVID-19 variants on the performance of an algorithm trained on specimens from 2020, 2021, and 2022. In other words, are new variants a concern for the long-term performance of MALDI-TOF-MS based identifications?
- Can the authors comment separately on the impact of (1) increased training set size and (2) the geographic diversity of combined datasets on the overall performance of the re-trained algorithm? In other words, can any of the improved performance of the re-trained algorithm be attributed to having a larger training set size (from the South American and Kazakhstan dataset)?
- The authors state that MALDI-TOF MS machine learning may be utilized in early stages of endemics/pandemics. Can the authors comment on the importance of the availability of characterized datasets for training such algorithms?

Recommend changes to tables/figures:

- Figure 1.
 - Ensure that appropriate BioRender license is obtained prior to publication and cited appropriately.
 - Caption: Please define “NPS” in the figure caption.
- Figure 2 and 3. Please increase font size so that it is legible.
- Table 1. Please define which values are included in parentheses in the table.
- In the main text, the authors may consider including a table that compares the performance of the original model with the re-trained model.

Minor recommendations:

Definitions:

- Throughout, consider referring to MALDI-MS as MALDI-TOF-MS
- Line 68. Please define in NP swabs were in a transport buffer
- Line 116. Define “PCA”
- Line 191. Please define “ROC”

Please include additional citations for R packages:

- Line 199. Include citation for ggtree and ggtreeExtra
- Line 116. Cite R FactoMinR and facoextra
- Line 136. Cite Caret R

Minor grammatical errors:

- Line 95. Missing space between “a” and “18-20 kV”

- Line 155: Italicize “et al”

We are grateful to both reviewers for their comments and feedback on our manuscript. Please kindly find our point-by-point responses to the reviewer's remarks below.

Reviewer #1 (Comments for the Author):

Application of MALDI-MS and Machine Learning to Detection of SARS-CoV-2 and non-SARS-CoV-2 Respiratory Infections' by Yegorov, et al. is a study about the practical use of MALDI and machine learning to differentiate between respiratory infections caused by SARS-CoV-2 versus those that are not, with the potential to use this technology in a clinical setting.

Discussion points

You point out that an advantage of this technique could be for limited resource labs in the early stages of a pandemic (Line 233), but how would that be possible without the large pool of samples to train the ML models on, and also with a standard of care test to show true positives?

Authors' response: The reviewer raises a very valid point. Our MALDI-TOF MS approach does not target specific viral structures but relies primarily on mass spectrometric signatures associated with ARI-induced perturbations in the nasal mucosa. This feature of the method makes it potentially valuable at the early stages of a pandemic when other tests (e.g. pathogen-specific molecular assays) may yet be unavailable.

However, as the reviewer rightly points out, further implementation of our approach would require access to sufficiently large training datasets, which is an important limitation to keep in mind. We have now highlighted these points in the Discussion (pp 9-10, lines 223-231 and p11, lines 258-260).

• Line 236, 254 - you acknowledge the importance of taking geographical location of the sample into consideration, so how would this technology work in the clinical lab? Would there be other options for testing if someone was from or had traveled out of the region when infected? How often would the ML model need to be recalibrated to ensure it wouldn't miss mutations/new strains? Would it be part of a testing algorithm?

Authors' response: Thank you for raising this question. We agree with the reviewer that the ML models would need to be recalibrated frequently to account for any changes to the ARI landscape. Again, as mentioned in our response above, the nature of our approach (which does not target specific pathogens but assesses molecular changes in the nasal mucosal environment) would make in theory relatively robust to changes in pathogen strains. We have now incorporated these points into the Discussion (pp 9-10, lines 227-231, and the Limitations section).

Line 73. Remove 'among'

Authors' response: Thank you- done!

Line 83. How long were samples frozen for before testing? Was there only one freeze/thaw?

Authors' response: Thank you for this question. Samples were collected and stored at -80C over the course of the study (2020-2022), and subsequently processed in batches once all samples have been collected. There was only one freeze/thaw cycle prior to MALDI-TOF MS. We have highlighted this in the Methods (p4, lines 87-88).

Table 1. Discrepancies in how percentages are written - some with a '.' and some with a ',' - I would change the three percentages that have a ',' (SARS-CoV-2+ column) to having a '.' for consistency.

Authors' response: Thank you for noting this discrepancy- it has been corrected.

Reviewer #2 (Comments for the Author):

The authors present an interesting extension of prior MALDI-TOF-MS and Machine Learning (ML) methods for the identification of COVID-19 and other non-COVID respiratory viruses in Kazakhstan. This work specifically highlights the importance of training ML algorithms on geographically diverse datasets. Overall, the findings of the study are well supported. Below are a few recommendations for expanded discussion, updates to figures/tables to improve clarity, and other minor changes.

Authors' response: We are thankful for the reviewers' positive view of our work! Below, we provide our detailed responses to each of the reviewer's comments.

Main points

In the discussion, the authors may consider commenting on the potential impact of COVID-19 variants on the performance of an algorithm trained on specimens from 2020, 2021, and 2022. In other words, are new variants a concern for the long-term performance of MALDI-TOF-MS based identifications?

Authors' response: Thank you for this question! We have now added this point to the discussion (p10, lines 227-231). As also mentioned in our response to Reviewer #1, our MALDI-TOF MS approach does not target specific viral structures but relies primarily on mass spectrometric signatures associated with ARI-induced perturbations in the nasal mucosa. Therefore, we believe that it would be more tolerant to changes in the circulating viral strains compared to viral structure-specific methods.

Can the authors comment separately on the impact of (1) increased training set size and (2) the geographic diversity of combined datasets on the overall performance of the retrained algorithm? In other words, can any of the improved performance of the retrained algorithm be attributed to having a larger training set size (from the South American and Kazakhstan dataset)?

Authors' response: Thank you for this question. We do believe that the improved performance of the retrained algorithm is due to both larger and more diverse training dataset. We have now highlighted this point in the Discussion (p10, lines 239-250).

The authors state that MALDI-TOF MS machine learning may be utilized in early stages of endemics/pandemics. Can the authors comment on the importance of the availability of characterized datasets for training such algorithms?

Authors' response: Thank you for pointing this out. The availability of training datasets in the early stages of an epi/pandemic would be a potential bottleneck to implementing our approach in practice. We have now added this point to the discussion (p 10, lines 232-238) and to the limitations section.

Recommend changes to tables/figures:

Figure 1. Ensure that appropriate BioRender license is obtained prior to publication and cited appropriately.

Authors' response: Thank you, we have now included the Biorender citation in the Acknowledgements.

Caption: Please define “NPS” in the figure caption. Figure 2 and 3. Please increase font size so that it is legible.

Authors' response: Thank you- done!

• Table 1. Please define which values are included in parentheses in the table.

Authors' response: Thank you- done!

In the main text, the authors may consider including a table that compares the performance of the original model with the re-trained model.

Authors' response: Thank you for this suggestion. We have now included Table 2 in the manuscript.

Minor recommendations:

Definitions:

Throughout, consider referring to MALDI-MS as MALDI-TOF-MS

Authors' response: Thank you- done!

Line 68. Please define in NP swabs were in a transport buffer

Authors' response: Thank you, this point has now been clarified in the Methods (p 3, lines 72-73)

• Line 116. Define “PCA”

Authors' response: Thank you- done!

• Line 191. Please define “ROC”

Authors' response: Thank you- done!

• Please include additional citations for R packages: Line 199.

Authors' response: Thank you- done!

Include citation for ggtree and ggtreeExtra

Authors' response: Thank you- done!

Line 116. Cite R FactoMinR and facoextra

Authors' response: Thank you- done!

Line 136. Cite Caret R

Authors' response: Thank you- done!

· **Minor grammatical errors:**

Line 95. Missing space between “a” and “18-20 kV”

Authors' response: Thank you- this has been corrected.

· **Line 155: Italicize “et al”**

Authors' response: Thank you, done!

Re: Spectrum04068-23R1 (Application of MALDI-MS and Machine Learning to Detection of SARS-CoV-2 and non-SARS-CoV-2 Respiratory Infections.)

Dear Dr. Irina Kadyrova:

Your manuscript has been accepted, and I am forwarding it to the ASM production staff for publication. Your paper will first be checked to make sure all elements meet the technical requirements. ASM staff will contact you if anything needs to be revised before copyediting and production can begin. Otherwise, you will be notified when your proofs are ready to be viewed.

Sincerely,
Heba Mostafa
Editor
Microbiology Spectrum

Reviewer #2 (Comments for the Author):

The reviewers have sufficiently addressed all comments.